# Detailed analysis of distorted retinal and its interaction with surrounding residues in the K intermediate of bacteriorhodopsin

Shoun Taguchi[1], Satomi Niwa [1], Hoang-Anh Dao[1], Yoshihiro Tanaka[1], Ryota Takeda [1], Shuya Fukai [1], Kazuya Hasegawa [2] & Kazuki Takeda [1✉]

The K intermediate of proton pumping bacteriorhodopsin is the first intermediate generated after isomerization of retinal to the 13-*cis* form. Although various structures have been reported for the K intermediate until now, these differ from each other, especially in terms of the conformation of the retinal chromophore and its interaction with surrounding residues. We report here an accurate X-ray crystallographic analysis of the K structure. The polyene chain of 13-*cis* retinal is observed to be S-shaped. The side chain of Lys216, which is covalently bound to retinal via the Schiff-base linkage, interacts with residues, Asp85 and Thr89. In addition, the Nζ-H of the protonated Schiff-base linkage interacts with a residue, Asp212 and a water molecule, W402. Based on quantum chemical calculations for this K structure, we examine the stabilizing factors of distorted conformation of retinal and propose a relaxation manner to the next L intermediate.

[1] Department of Chemistry, Graduate School of Science, Kyoto University, Kyoto, Sakyo-ku 606-8502, Japan. [2] Structural Biology Division, Japan Synchrotron Radiation Research Institute (JASRI), 1-1-1 Kouto, Sayo-cho, Sayo-gun, Hyogo 679-5198, Japan. ✉email: ktakeda@kuchem.kyoto-u.ac.jp

Bacteriorhodopsin (bR) is a light-driven proton pump occurring in the plasma membrane of an archaeon, *Halobacterium salinarum*[1,2], and is currently one of the most intensively studied membrane proteins[3,4]. Similar microbial rhodopsins acting as ion pumps, channels, and sensors have been discovered in not only archaea but also bacteria, fungi, and even viruses[5–7]. All of these proteins, like bR[8], have seven transmembrane helices and a central retinal chromophore. Retinal has a polyene backbone with long conjugated double bonds. In the case of bR, the retinal chromophore forms a Schiff-base (SB) linkage with a lysine residue at position 216 (Lys216). Retinal takes an all-*trans* conformation in the light-adapted resting state. The light absorption causes the isomerization of a double bond at C13−C14 of retinal from a *trans* to *cis* conformation. The structural changes are propagated to the protein portion and the respective functions are expressed.

In the case of bR, a reaction cycle occurs in which a series of intermediates (I, J, K, L, M, N, O) are sequentially generated. In the reaction cycle of bR, a single proton is transported from the inside to the outside of the plasma membrane. The K intermediate, which is characterized by an absorption peak at 590 nm at room temperature[9], is the first intermediate generated after isomerization of retinal to the 13-*cis* form. The K intermediate can be trapped by the light irradiation under cryogenic temperatures[10]. The K intermediate produced at cryogenic temperatures has been shown to have almost the same characteristics in vibrational bands as those produced transiently at room temperature[11,12]. The structure of the K intermediate was first investigated by means of the cryo-trapping technique in combination with X-ray crystallography[13–16] (Supplementary Table 1). The structure was also investigated by means of time-resolved serial femtosecond crystallographic (TR-SFX) studies[17–19] (Supplementary Table 1). These investigations revealed that the structural changes upon the formation of the K intermediate are small and still limited to the vicinity around retinal. Therefore, the K intermediate is an interesting research subject with the potential to provide insight into the storage and propagation of energy within proteins. However, many inconsistencies can be found among the previously reported K structures in the conformation of retinal and the details of its interaction with surrounding residues. For this reason, the crucial points in the storage and propagation of energy within proteins remain poorly understood, although an estimation of the stored energy has been made ~50 kJ/mol[20].

The ability to manipulate cells by light has recently led to the use of light-driven rhodopsins for the deliberate activation of neuron and other cells by optogenetics[21,22]. In addition, extensive studies have been performed on the use of bR in photo materials[23,24]. In terms of the development of applications using rhodopsins and other photoactive proteins, it is important to elucidate the mechanism by which photoisomerization of the chromophore is translated into functional movements in proteins. Therefore, in this study, we focused on bR, which is one of the best-studied photoactive proteins, and performed high-resolution structural analysis to accurately determine the conformation of its retinal chromophore and the interactions with the protein surroundings in the K intermediate.

## Results

**X-ray diffraction data collection and data analysis**. To accumulate the K intermediate, we employed a cryo-trapping procedure[15] in which a light-adapted bR crystal was irradiated with a green light at 100 K (Fig. 1a). Diffraction data from a crystal including the K intermediate at ~1.3 Å were collected at 15 K with an absorbed dose of 0.05 MGy. This absorption dose value was one-third of the limit at 15 K[25]. Subsequently, diffraction data of the ground state were collected from the same crystal after irradiation of red light. The average $I/\sigma(I)$ value for the highest shell (1.42–1.33 Å) was a sufficiently high value of ~1.4, when the resolution limit was determined as a shell with $CC_{1/2}$ of ~0.5 (Table 1). As in previous studies, significant densities were observed only around retinal in the difference Fourier map (Fig. 1b). On the other hand, difference electron densities were not observed at the regions away from retinal, such as the proton uptake site, proton release site, or surface of the protein. The strongest positive and negative densities were observed around the SB linkage and the C20 atom of retinal (Fig. 1c). In addition, pairs of relatively strong densities were observed on the C14 and C15 atoms. Relatively strong densities were also observed corresponding to changes in individual atoms of the other portions of retinal, the side chain of Lys216, and surrounding residues. Therefore, it was possible to determine the structure of the K intermediate unambiguously (Supplementary Fig. 1).

We were able to collect data at the same level of resolution from another crystal (Supplementary Table 2). Using the data, we obtained an almost identical difference Fourier map (Supplementary Fig. 2a, b). Moreover, when the ground state data were first collected prior to the K state data, the difference Fourier map showed almost the same densities as in the above cases (Supplementary Fig. 2c, d). This fact indicates that the dependence on the sequence of measurement is not so large for the structural changes between the bR and K, as suggested from spectroscopic measurements[26]. It also shows that the X-rays exerted no damaging effects at a total dose of ~0.1 MGy. The three K structures are indeed to be almost identical according to the superimposition (Supplementary Fig. 3), although only the shape of the EF loop in crystal II differ from other two. This may be due to the difference in the lattice constant of the *c*-axis (Supplementary Table 2). In fact, the root mean square deviation (rmsd) bwteen K in crystal II and K in crystal I is 0.9 Å, larger than 0.4 Å of rmsd between K in crystal III and K in crystal I.

As for the ground state, X-ray diffraction data from crystal I and II were measured after the activation with green laser and subsequent restoration with red laser, whereas data from crystal III were measured before the activation. Nevertheless, three structures are also nearly identical as in the case of the K structures (Supplementary Fig. 4a-c). This implies that the process of activation does not affect the ground state structure. This implication is also supported from the fact that the difference Fourier map between data of crystal I and crystal III shows no significant densities around retinal and the proton channel (Supplementary Fig. 4d-f).

**Distortion in 13-*cis* retinal**. By fitting the structure of retinal to the electron density with high resolution, it was possible to determine the conformation of retinal with high accuracy. The standard deviations of the dihedral angles of the polyene chain was ~10° in the K structure. The polyene chain bends largely to the helix F side near the C14 atom, and the C13−C20 bond leans to the Tyr185 side of helix F (Fig. 2a). On the other hand, the polyene chain bends against helix B near the C9 atom. As a result, the 13-*cis* retinal in the K intermediate is curved in an S-shape when viewed from the cytoplasmic side. This retinal conformation could provide a very good explanation of the difference electron densities on the difference Fourier map. When the torsion angles were compared between bR and K, the most significant difference was found in the C13−C14 bond occurring the *trans–cis* isomerization (Fig. 2b). However, the difference was ~120°, not 180°. Due to changes in other bonds, the orientation of

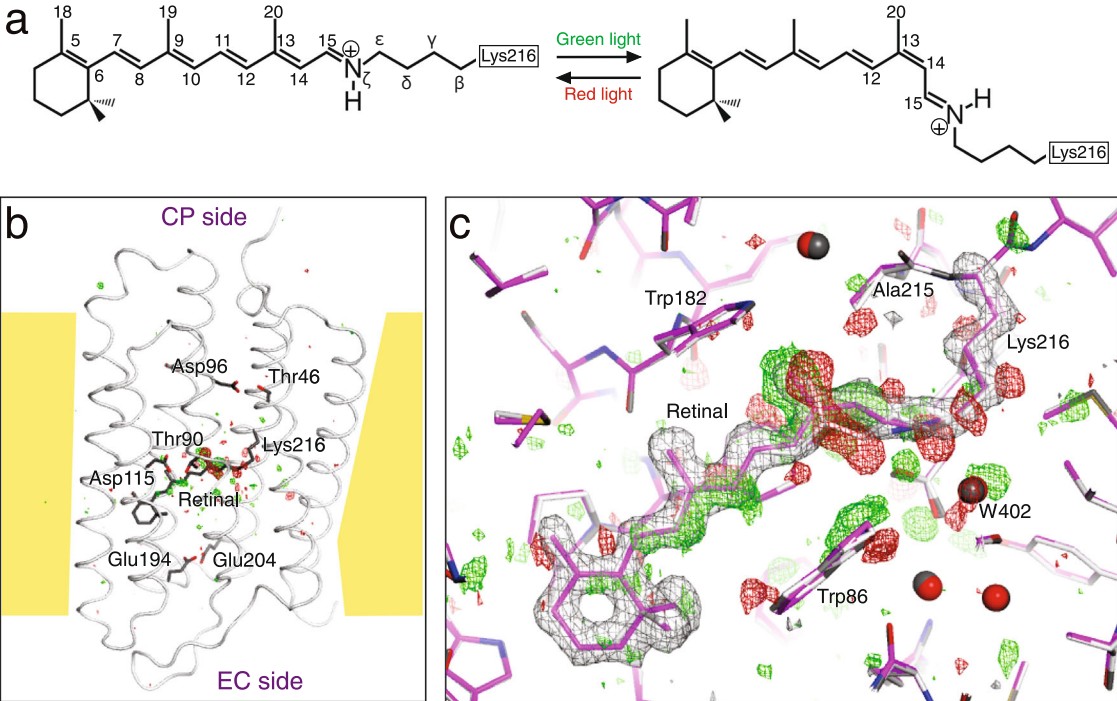

**Fig. 1 Structural analysis of the K intermediate of bR. a** Photoisomerization of retinal. All-*trans* retinal, which is linked with Lys216 of bR *via* the SB linkage, isomerizes to 13-*cis* retinal by absorption of green light. A back reaction is caused by absorption of red light. **b** A whole view of the $F(K + bR) - F(bR)$ difference map. The difference electron densities at the $+4\sigma$ and $-4\sigma$ levels are shown as green and red meshes, respectively. The structure of bR is represented as a tube in gray, while retinal is shown as sticks in dark gray. **c** A close-up view around the retinal chromophore. The K structure is represented as colored sticks, while the ground state structure is represented as gray sticks. The $F(K + bR) - F(bR)$ difference map is shown as green and red meshes at contour levels of ±3σ. The omit map for retinal and the Lys216 side chain calculated from the extrapolated data is overlaid as gray meshes at a contour level of $+3\sigma$.

the Nζ−H bond in K finally changes by only 24° from that in the ground state. This small relative change was consistent with a previous polarized FTIR result of ~25°, whereas the absolute angles from the membrane normal did not agree[27]. The next largest difference in the polyene chain was found in a torsion angle of the C15−Nζ bond. In addition, small significant changes were found in most of the other torsion angles in the polyene chain of retinal. The difference Fourier map indicated that the position of the β-ionone ring shifts slightly to the SB direction due to the S-shaped curve of the polyene (Fig. 1c). As for the side chain of Lys216, which forms the SB linkage with retinal, the torsion angles for the Cε−Nζ and Cδ−Cε bonds exhibited no significant deviations from the ground state despite the single bond, indicating the stabilization by interactions with the surrounding residues.

**Propagation of structural changes.** Although the structural changes are limited to the vicinity of retinal, atomic details of the structural change upon the formation of the K intermediate could clearly be observed in the surrounding residues. For the side chain of Lys216, Cε, and Cδ move toward helix G in addition to Nζ of SB (Fig. 3a). Weak difference densities were observed along the main chain of residues in helix G from Phe208 to Gly218. By the tilting of the C13−C20 bond, Tyr185 moves toward Leu211, inducing the movement of Leu211 (Fig. 2a and Fig. 3b). The side chain of Trp86 rotates around the Cα−Cβ axis, filling the concave portion of retinal generated by the *trans-cis* isomerization (Fig. 3b). The side chains of Arg82 and Phe208 are also shifted by the movement. Ser141 in helix E and Pro186 in helix F move toward the ionone ring of retinal (Fig. 3c). In addition, several residues belonging to helix E (Ala139−Thr142) and helix F

(Leu181−Val187) also move with them slightly. The structural changes in the main chains of the helices E, F, and G may be related to peptide bands insensitive to the H/D exchange observed in the difference FTIR spectra[28].

The hydrogen bond distance between Asp85 and Thr89 is only slightly shortened even by movement of the side chain of Thr89 toward the retinal upon the formation of the K intermediate (Fig. 3d). The hydrogen bond distance between Asp85 and W402 increases compared to the ground state structure. On the other hand, the distance between Asp212 and W402 is shortened to 3.08 Å, although the distance in the ground state is 3.40 Å (Fig. 3e). The distance between Lys216 and Asp212 is also shortened from 3.93 Å for the ground state to 3.27 Å for K by the rotation around the Cβ−Cγ bond of Asp212 in addition to the isomerization of retinal. It should be noted that a similar rotation at Asp212 was observed in earlier I and J intermediates, but not in K, in a TR-SFX study[18]. A protonated aspartate, Asp115, in the vicinity of the retinal seems to have no direct involvement in the proton pumping. Asp115 forms two hydrogen bonds with Thr90, both in the ground state and the K intermediate (Fig. 3f). However, the manner of hydrogen bonding changes slightly by moving of Asp115. This may correspond to a change in the environment of Asp115 suggested by a previous FTIR result[29].

**Changes in the interaction of retinal and surrounding residues.** Due to the large positional changes of C20 by the isomerization of retinal, the distance between C20 and Cε1 of Tyr185 in helix F is decreased to 3.45 Å from 4.25 Å in the ground state (Fig. 4a and Supplementary Fig. 5a). However, the distances between C13 and atoms of Tyr185 are not largely changed. As for the manner of interaction between C20 and Trp182 located on the cytoplasmic

**Table 1 Data collection and refinement statistics.**

|  | bR + K | bR | K_ext |
|---|---|---|---|
| *Data collection* |  |  |  |
| Accumulated dose (MGy/position) | 0.05 | 0.10 |  |
| Space group | $P6_3$ | $P6_3$ |  |
| Cell dimensions, $a$, $c$ (Å) | 60.57, 110.88 | 60.57, 110.89 | 60.57, 110.88 |
| Resolution (Å) | 50−1.33 (1.35-1.33)[a] | 50−1.33 (1.35-1.33) | − |
| $R_{sym}$ (%) | 7.4 (115.7) | 7.4 (116.2) | − |
| $I/\sigma(I)$ | 11.1 (1.4) | 11.0 (1.3) | − |
| Completeness (%) | 99.9 (99.8) | 99.9 (99.7) | − |
| Redundancy | 5.8 (5.5) | 5.8 (5.5) | − |
| $CC_{1/2}$ (%) | 99.8 (46.4) | 99.8 (45.6) | − |
| *Refinement* |  |  |  |
| Resolution (Å) | 50−1.33 | 50−1.33 | 20−1.33 |
| Total reflections | 52170 | 52290 | 52152 |
| Twin ratio | 0.20 | 0.19 | 0.21 |
| Fraction of K | 0.19 | 0.0 | 1.0 |
| $R_{work}/R_{free}$ (%) | 13.1/17.3 | 13.6/17.6 | 27.4/29.3 |
| No. of atoms | (bR/K) |  |  |
| Protein | 1855/1779 | 1855 | 1820 |
| Retinal | 20/20 | 20 | 20 |
| Lipid | 374/0 | 374 | 220 |
| Water | 64/21 | 64 | 26 |
| *No. of multi-conformations* | 10/0 | 10 | 5 |
| *B-factors* | (bR/K) |  |  |
| Protein (Å²) | 24.9/25.3 | 26.2 | 27.3 |
| Retinal (Å²) | 15.9/15.7 | 16.3 | 16.9 |
| Lipid (Å²) | 66.6/− | 63.5 | 68.2 |
| Water (Å²) | 37.1/32.2 | 41.5 | 26.7 |
| R.m.s. deviations |  |  |  |
| Bond lengths (Å) | 0.024 | 0.014 | 0.009 |
| Bond angles (°) | 2.0 | 2.3 | 2.1 |

[a]Values for the highest resolution shell are in parentheses.

side of retinal, the distance to Cδ1 of Trp182 is decreased to 3.22 Å from 3.76 Å, while the distance between C20 and Nε1 of Trp182 is almost unchanged. Energetically unfavorable interactions with short distances are not observed in our K structure. Therefore, it can be assumed that CH⋯π and π⋯π interactions observed between retinal and these surrounding residues are attractive interactions.

For hydrogen bonding on the EC side of the SB linkage, only Nζ of Lys216 shows significant changes upon the K formation (Fig. 4b and Supplementary Fig. 5b). Although the distance between Nζ of Lys216 and Oδ1 of Asp212 in the ground state is 3.93 Å, indicating no significant interaction, it decreases to 3.27 Å in the K intermediate. The distance between Nζ of Lys216 and W402 increases by 0.28 Å to 3.07 Å. The σ levels of the differences for these two distances were larger than 6. On the other hand, the values for the other hydrogen bonds were lower than 3, indicating that the changes of distances in these hydrogen bonds are negligible levels (Supplementary Table 3).

**Comparison with other K structures**. Until now, three and two X-ray structures determined with the cryo-trapping and TR-SFX techniques, correspondingly, have been reported for the K intermediate[13–19]. Although the retinal conformation in most structures took distorted conformations as predicted by the spectroscopic data[30], these data showed a large variation in the crystallographic structures (Fig. 5a). The degree of difference

seemed not to depend on the method used to produce the K intermediate. The largest discrepancies were found in the positions of the C13, C14, and C20 atoms of retinal. As a result, among the various structures, the variation was ~60° for C13−C14 and C15−Nζ, while the variations for the other torsion angles were ~40° for the torsion angles of the retinal polyene chain (Supplementary Fig. 6). These variations were significantly larger than those for the ground state, indicating the absence of reliable structural parameters for the K intermediate (Fig. 5b). In our structure, the torsion angles of C13−C14 and C15−Nζ were −38° and 143° respectively, and the standard deviations calculated from three structures (Crystals I—III) were 14° and 11°, respectively (Supplementary Table 4). These values were at the center of other results in the plot (Fig. 5b). The closest structures in the torsion angle plot were 6G7K[18] and 1M0K[14]. Although the positions of the C13, C14, and C20 atoms in 6G7K[18] were close to those in our results, those in 1M0K[14] were quite different. This difference was caused by the planarity of the polyene chain between C9 and C13. These deviations from 180° also contributed to the curvature to the polyene chain, resulting in the S-shaped conformation.

Due to the variety in the conformation of retinal, significant differences were also observed in the interactions between retinal and the surrounding residues, such as Trp182 and Tyr185 (Supplementary Fig. 7). One of the K structures (1QK0)[13] exhibited the most different interaction manner in comparison to that of our structure (Supplementary Fig. 7a). On the other hand, another K structure (6G7K)[18] appeared to be the most similar to ours in terms of the interactions around retinal, while some uncommonly close distances were found between retinal and Tyr185 (Supplementary Fig. 7d). In addition, a short distance of 2.28 Å was also observed in the hydrogen bond between Tyr185 and Asp212. Because the positions of atoms in retinal were very similar, the difference from our K structure was due to the degree of structural changes in Tyr185 (Fig. 4a and Supplementary Fig. 7d). As for 7Z0C[16], the crystal size, experimental conditions, and age of measurement are very similar to those of 7XJC in this study. Nevertheless, there are non-negligible differences at the positions of the C13, C14, and C20 atoms in retinal (Fig. 5a). This is probably due to the difference in the torsion angle of the single Cε−Nζ bond, rather than the difference in the torsion angles of the conjugated double bonds in the polyene chain of retinal. Consequently, a marked difference resulted in the orientation of the Nζ−H. These may also be a reason for significant differences in the interaction with Tyr185 (Fig. 4a and Supplementary Fig. 7f). A possible reason for the difference could be the number of repetitions of the bR↔K change during the measurement of the diffraction data. In the diffraction data collection of 7XJC, the diffraction data of K + bR irradiated with green light was measured at first in order to obtain the K structure as free from X-ray damage as possible, as described above. On the other hand, in the data collection of 7Z0C, the bR↔K was repeated many times with changing angles[19].

**NCI analysis of interactions around the SB linkage**. The hydrogen atom-complemented structure was obtained by means of QM/MM calculations starting from our K structure in order to perform detailed investigations of interactions around the SB linkage. The optimized structure was examined by the non-covalent interaction (NCI) plot which is an isosurface of the reduced density gradient calculated from an electron density $\rho$[31,32]. There was a bluish-green surface in the NCI plot between the amide-proton on Nζ of Lys216, which forms the SB linkage, and Oδ1 of Asp212, indicating the presence of a

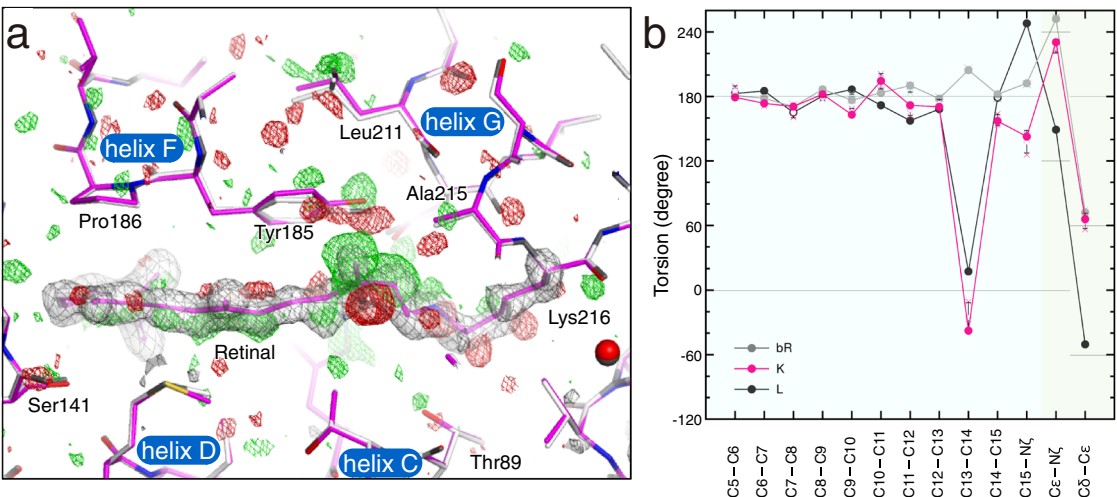

**Fig. 2 Details of the distorted retinal conformation. a** S-shaped polyene backbone of retinal viewed from an upper part of Fig. 1b. The maps are shown in the same manner as in Fig. 1b. **b** Torsion angles along the retinal polyene chain in the K structure (7XJC) are plotted in magenta. The standard deviations, which were calculated from three structures, are indicated as error bars on the mean values shown as hollow circles. The values for respective structures are shown as crosses. The values can be found in Supplementary Table 4. Those for the ground state structure in this work (7XJD) and a recent L structure[16] are plotted in gray and black, respectively. The black horizontal lines indicate angles with local minimum energies for the respective torsions.

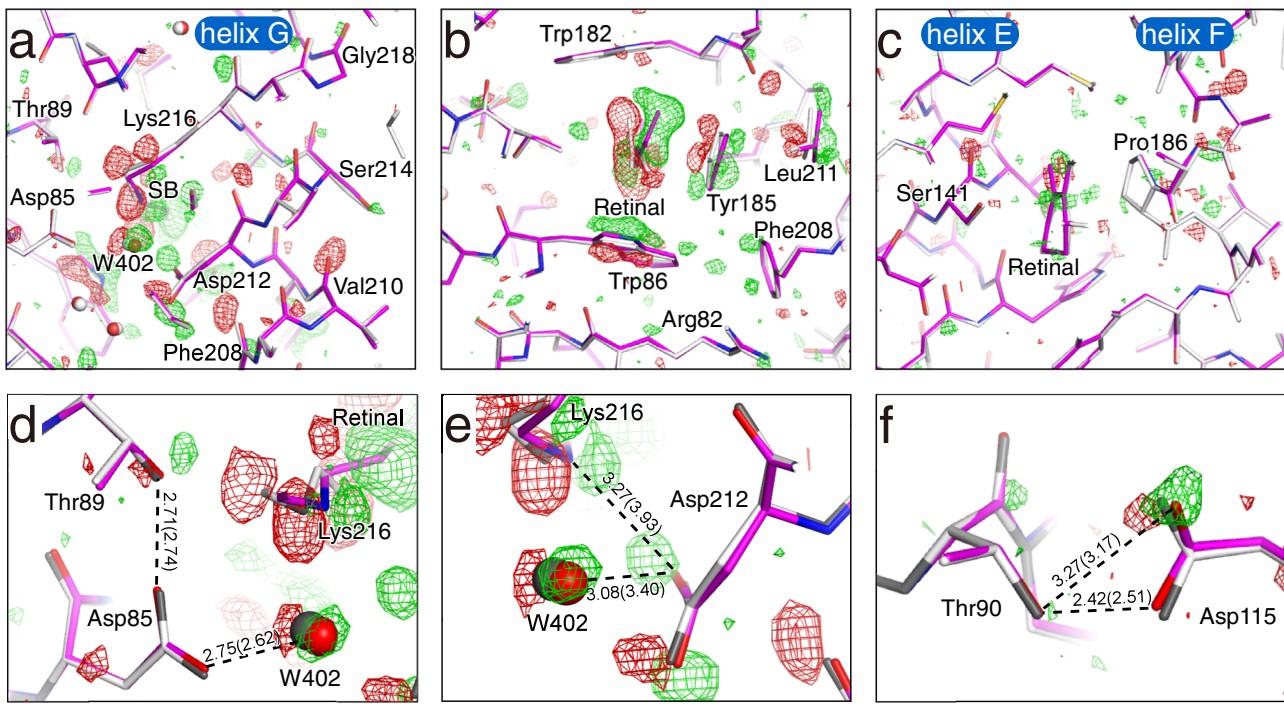

**Fig. 3 Propagation of structural changes from retinal to the protein matrix. a** A cross-section of retinal at the SB linkage. The $F(\text{K} + \text{bR}) - F(\text{bR})$ difference map is shown as green and red meshes at contour levels of ±3σ. The K structures are represented as colored sticks. The ground state structure is overlaid as gray sticks. **b** A cross-section of retinal at the middle of the polyene backbone. **c** A cross-section of retinal at the β-ionone ring. **d** The map around Asp85. **e** The map around Asp212. **f** The map around Asp115.

relatively strong interaction (Fig. 6a). This interaction may have been further enhanced by the electrostatic attraction between the positive charge at Nζ of Lys216 and the negative charge at Oδ1 of Asp212. In addition, the interaction between Nζ of Lys216 and W402 was also observed as a bluish-green surface. On the other hand, there were green surfaces between the methylene hydrogen atoms of the Cε of Lys216 and the oxygen and hydrogen atoms of Thr89, indicating the presence of attractive van der Waals interactions. In addition, a green NCI surface was observed between the Cε of Lys216 and the Oδ1 of Asp85, suggesting CH···O type hydrogen bonding[33]. Thus, the rotation of the Cε−Nζ and Cδ−Cε bonds was suggested to be inhibited by these interactions between the Cε and Nζ of Lys216 and the surrounding residues (Fig. 6b). In fact, the polyene chain portion of the retinal in the K intermediate showed deviations from the 180° or 0° which should take despite the double bond, while the side chain of Lys216 showed little deviation despite the single bonds (Fig. 2b).

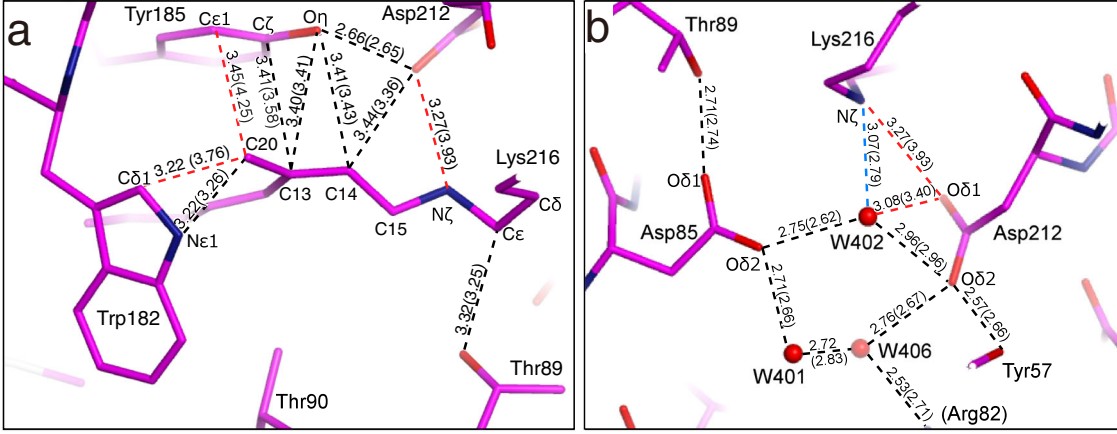

**Fig. 4 Details of interactions in the K intermediate. a** Interatomic distances of <3.5 Å between retinal and the surrounding residues are shown, while those for the ground state are in parenthesis. The dotted lines are colored according to the changes of the distances upon the K formation, with red and blue indicating a large (>0.2 Å) decrement and large increment, respectively. **b** Hydrogen bonds with distances of <3.5 Å on the EC side of the SB linkage are shown, while those for the ground state are in parenthesis.

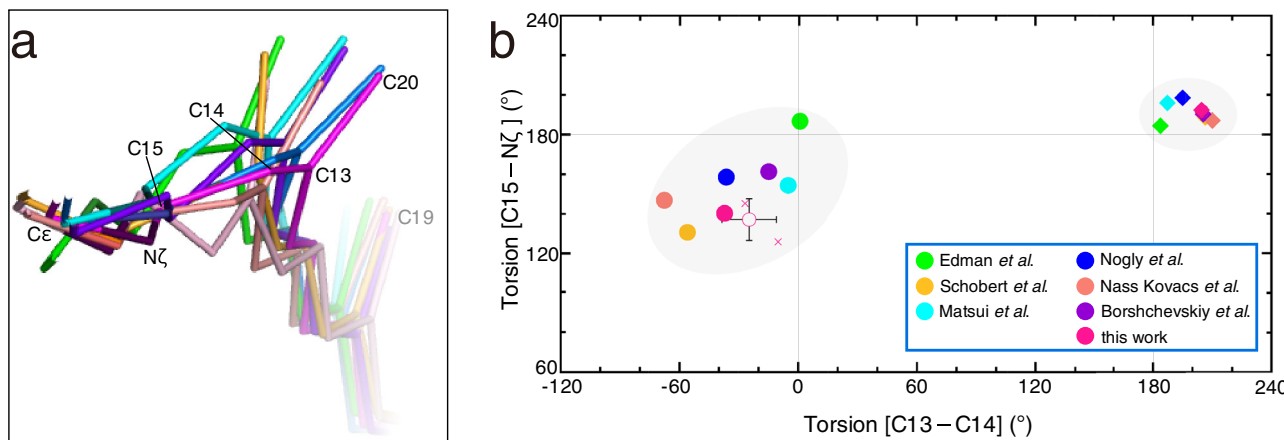

**Fig. 5 Comparison with other K structures. a** Superimposition of various K structures. Carbon atoms in 1QK0 reported by Edman et al.[13], 1M0K by Schobert et al.[14], 1IXF by Matsui et al.[15], 6G7K by Nogly et al.[18], and 6GA6 by Nass Kovacs et al.[19], 7Z0C by Borshchevskiy et al.[16] and 7XJC reported in this work are colored in green, yellow, cyan, navy orange and magenta, respectively, while nitrogen (blue) atoms are in common. **b** Distribution of torsion angles of conjugated double bonds (C13−C14 and C15−Nς) in retinal. The torsion angles for the K structures are plotted as filled circles in the same colors as used for the carbon atoms in panel a, while those for the corresponding ground state structures (1QKP[13], 1M0L[14], 1IW6[15], 6G7H[18], 6RMK[19], 7Z09[16] and 7XJD) are plotted as filled diamonds. For the K and ground state structures in this study, the standard deviations calculated from three structures for each are shown as error bars on the mean values shown as hollow circles. Data points for respective structures in this study are shown as crosses.

## Discussion

Although high-resolution structures of other states of bR have been reported recently[16,25], that of K has remained ambiguous despite the importance of the distorted structure in the proton transport mechanism. Based on the structural analysis of the K intermediate at ~1.3 Å in this study, however, we were able to obtain the accurate interaction manners as well as the conformation of the distorted 13-*cis* retinal. By comparing the retinal conformation in K in this study to that in a recent L structure[16], significant changes in the torsion angles of the C13−C14, C14−C15, and C15−Nς bonds were observed. This fact indicates that the twist in the polyene chain of the retinal in the K intermediate has been relaxed in the L intermediate. In addition to those changes, the Cε−Nζ and Cδ−Cε bonds rotated by ~120° to form another rotational isomer. Thus, the K intermediate was found to be stabilized by the interactions between Cε/Nζ and the surrounding residues. This provided insight into how the rotations around the Cε−Nζ and Cδ−Cε bonds occur during the transition to the L intermediate. We could thus interpret that the

sub-states of the K intermediate that have been suggested to exist are generated due to changes in the dihedral angle of the polyene chain of the retinal and the side chain of Lys216. Some sub-states, such as $K_0$, $K_E$, and $K_L$, have been proposed in the K intermediate[34–36]. Some of the various crystallographic K structures, which are different from each other as described above, may be such sub-states captured by slightly different experimental conditions (Supplementary Table 1). Therefore, an examination of the differences in each structure can provide information about the sub-states in the K intermediate. Based on only differences in the interaction between Nζ-H and W402 and/or Asp212 (Supplementary Fig. 8), the change near the SB can be proposed as $J \rightarrow 7XJC$ (this work) $\rightarrow 1M0K^{14} \rightarrow 6GA6^{19} \rightarrow 1IXF^{15} \rightarrow 6G7K^{18} \rightarrow 7Z0C^{16} \rightarrow 1QK0^{13} \rightarrow L$. However, it is not possible to clearly determine the correspondence between these and the sub-states proposed so far. In addition, the order cannot explain the changes in the distortion of retinal and the movement of Tyr185. Furthermore, some cautions are required in the discussion as follows. 1QK0[13] and 1M0K[14] were determined in

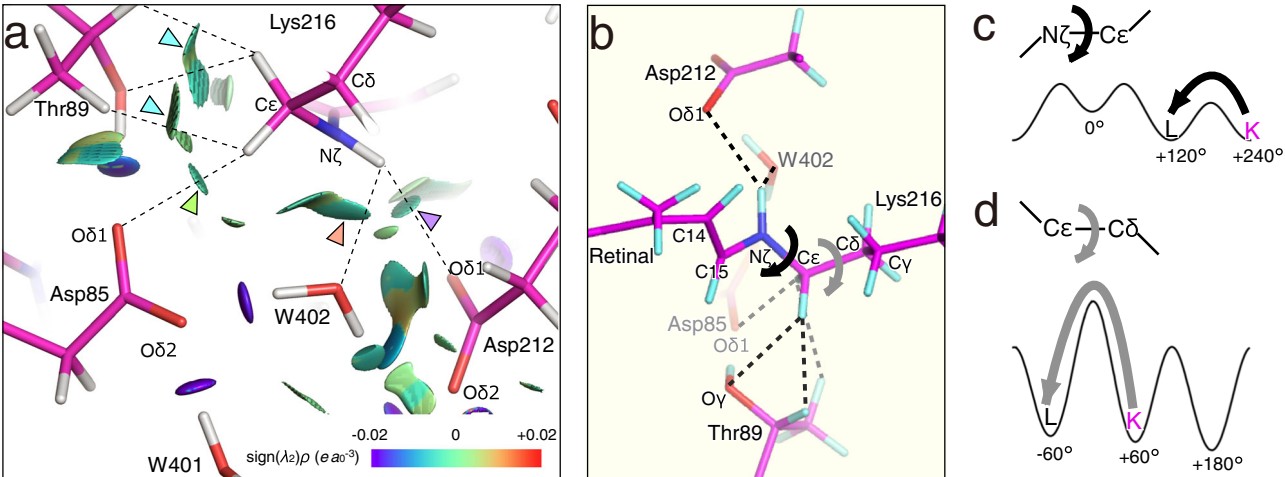

**Fig. 6 Interactions stabilizing the retinal conformation of K. a** The NCI plot for the EC side of the SB linkage. Hydrogen atoms are added according to the optimization with the QM/MM calculation. The reduced density gradient isosurface at $s(\rho) = 0.5$ is added as colored surfaces. The nature of the interaction can be visualized by the color according to the value of $\mathrm{sign}(\lambda_2)\rho$ in an atomic unit ($e\,a_0^{-3}$), where $\lambda_2$ and $\mathrm{sign}(\lambda_2)$ are the second eigenvalue of the Hesse matrix of $\rho$ and its sign, respectively. The color scale are given at the right bottom of the panel. Blue, green and red correspond to attraction, weak attraction and repulsion, respectively. The blue arrows indicate the interaction between H atoms of Cε in Lys216 and Thr89. The green arrow indicates the interaction between H of Cε in Lys216 and Asp85. The orange arrow indicates the interaction between H of Nς in Lys216 and W402. The purple arrow indicates the interaction between H of Nς in Lys216 and Asp212. **b** Interactions with surrounding residues (Asp85, Thr89 and Asp212) and Water402 prevent torsional rotation at Nς–Cε and Cε–Cδ bonds in Lys216. **c** The torsional potential energy curve for the Nς–Cε bond. The torsion angles in K/bR and L are ~+240° and +120°, respectively. **d** The torsional potential energy curve for the Cε–Cδ bond. The torsion angles in K/bR and L are approximately +60° and −60°, respectively.

analyses where the influences of X-ray damage were not sufficiently taken into account. 1IXF[15] has a lower resolution than the others, despite the fact that efforts have been made to examine and remove the effects of X-ray damage. For 1QK0[13], however, there is a distinct difference from other K intermediates with respect to the angle of rotation around the Cε−Nζ bond (Supplementary Fig. 6). So, the possibility that 1QK0[13] is an experimental capture of a sub-state ($K_L$) proposed by a computational study[37] is highly plausible even in the presence of such problems. Thus, rotation of the Cε−Nζ bond may occur prior to rotation of the Cδ−Cε bond. This may correspond to the fact that the depth of the potential around the Cε−Nζ bond (~10 kJ/mol) is only about half that of the potential around the Cδ−Cε bond (~20 kJ/mol), due to the fact that Nζ has only one hydrogen atom (Fig. 6c, d).

There are two scenarios for the change in hydrogen bonding between Nζ and W402 at the rotation of the Cε−Nζ bond. The first is the case where this hydrogen bond is broken by rotation. In this case, even after the breaking of the hydrogen bond, the water molecule W402 may remain in the vicinity of its original position by hydrogen bonds with Asp85 and Asp212. In the second case, on the other hand, the hydrogen bond between Nζ and W402 is still formed after the rotation of the Cε−Nζ bond. W402 is consequently dragged into the CP side by the hydrogen bond as suggested by Kouyama et al.[38]. In this case, the cleavage of strong hydrogen bonds between W402 and Asp212 or Asp85 must occur at this time. In consideration of the rotation around the Cδ−Cε bond immediately following the rotation of the Cε−Nζ bond, the methylene hydrogen atoms of Cε may disturb these hydrogen bonds. The migration of W402 during the transition from K to L should be investigated further by molecular dynamics calculations with reference to our present findings.

There is no dispute that TR-SFX makes a great contribution to the study of protein conformational changes. However, the intense light pulses of pump lasers have been used as a trigger in the determination of reaction intermediates in TR-SFX[39]. It has

been pointed out that the contribution of multiphoton processes due to high photon densities can be problematic in such experiments[40]. In the case of bR, the influence of multiphoton absorption by Trp182 was seen as a problem[40]. The movement of water molecules far from the retinal, such as W401 and W406 in K or earlier intermediates observed in the TR-SFX results[18,19], may be influenced by this multiphoton absorption (Supplementary Fig. 8d, e). Therefore, unlike in the case of photoreaction intermediates that appear after a long time, such as the M intermediate, the effects of high photon flux of pump lasers must be taken into account in the case of the K intermediate, which appears immediately after photoisomerization. On the other hand, in the present study, the K intermediate structure was determined by the cryo-trapping method using lasers (~0.1 W/cm²) with light intensity comparable to that of sunlight, so this problem of multiphoton absorption did not exist. Therefore, the importance of the K intermediate must be carefully investigated by using structures from both the time-resolved and cryo-trapping methods. This study may provide a starting point for discussing the validity and limitations of TR-SFX results with high-intensity pump lasers. Before we can elucidate the detailed mechanism of the functional expression of photoactive proteins, further comprehensive research combined with computational methods[41] will also be necessary for the field of structural biology.

## Methods

**Preparation of crystals.** bR was purified from the purple membrane of *H. salinarum* strain R1 (JMC9409) by a standard protocol[1]. The crystals of bR used in this study were prepared by the LCP method as previously described[25]. In brief, 32 μL of the monoolein-based LCP matrix supplemented with 1.6% squalane (Sigma-Aldrich) and 8% trehalose C16 (Dojindo) was mixed with 20 μL of 15 mg/mL bR solution. Each 2 μL of the mixture was immersed in 40 μL of 2.0 M Na/K phosphate (pH 5.6) on a microbridge (Hampton Research, Aliso Viejo, CA), and equilibrated with 500 μL of 2.0–2.5 M Na/K phosphate in a 24-well crystallization plate. Crystals were harvested after 6−12 months. The excess LCP matrix surrounding the bR crystals was removed with squalane[42]. Immediately (within ~1 min) after light adaptation for more than 5 min with a white halogen lamp in addition to light from a microscope, bR crystals were frozen with a nitrogen gas flow of 100 K and stored in a liquid nitrogen tank.

**X-ray diffraction data collections**. Diffraction experiments were carried out at beamline BL41XU of SPring-8. The K intermediate of crystal I ($350 \times 350 \times 30\ \mu m^3$) was accumulated by irradiation of light from a green laser (532 nm, ~1 mW) for 5 min at 100 K. The crystal was occasionally rotated by 180° during the laser irradiation. The temperature was lowered to 15 K immediately after the laser irradiation. Diffraction data for the K intermediate were collected with an Eiger X 16 M detector (Dectris). The detector distance, X-ray wavelength, and beam size were set to 135 mm, 0.80 Å, $50 \times 20\ \mu m^2$, respectively. Diffraction data for 110° were collected with the helical data collection method in which the oscillation range and exposure time for one frame were 0.2° and 0.2 sec, respectively. Diffraction data of the ground state were subsequently collected by the same measurement conditions from the identical crystals at 15 K, after irradiation of light from a red laser (678 nm, ~1 mW) for 5 min at 100 K. The data for crystal II were collected in the same way as those for crystal I. Only for crystal III, the X-ray diffraction data of the ground state were initially collected prior to the data of the K intermediate. The X-ray absorption dose was estimated with the program RADDOSE[43]. The dose was estimated to be 0.05 MGy per one dataset, and 0.1 MGy in total of two datasets. The total dose is lower than the dose limit at 15 K of 0.15 MGy estimated in our previous study[25].

**Crystallographic analysis**. All diffraction data were processed with the program XDS[44]. At the beginning of the structure analysis, the ground state structure was refined for each crystal using 5ZIL as an initial model[25]. The accumulation of the K intermediate was first checked with the $F_o(K + bR) - F_o(bR)$ difference Fourier map using the phases from the ground state structure. $F$ (K) and $\sigma(F$ (K)) were extrapolated using the equations[45]

$$|F_K| = \left(1 - \frac{1}{f}\right)|F_{bR}| + \frac{1}{f}|F_{bR+K}| \tag{1}$$

$$\sigma(F_K) = \sqrt{\left(\left(1 - \frac{1}{f}\right)\sigma(F_{bR})\right)^2 + \left(\frac{1}{f}\sigma(F_{bR+K})\right)^2} \tag{2}$$

where $f$ is the fraction ratio of the K intermediate. The $f$ value was determined judging from the shape features of the retinal portion on the $F_o(K) - F_c(K)$ omit map and temperature factors of retinal after the refinement of the extrapolated datasets with various $f$ values using the program CNS[46]. In the refinement calculations, only the coordinates of retinal and residues with densities on the $F_o(K + bR) - F_o(bR)$ difference Fourier map were moved. The structures were corrected with the program COOT[47]. The K structure of the most plausible $f$ value was subsequently refined against the extrapolated dataset with the program SHELX[48]. The K structure was used in the subsequent SHELX refinement against $F(K + bR)$, where the K and ground states were treated as double conformations. As for the twin fraction, the initial value was estimated with the L test[49] with the program CTRUNCATE in the CCP4 program suite[50]. The twin fraction was optimized through the refinement calculations for the ground state and the extrapolated K structures. On the other hand, the twin fraction for the K + bR structure was optimized only in some final steps of the SHELX refinement. Twined data are processed using the method of Pratt et al. and Jameson in the SHELX program[51–53]. The refined structure was checked with the program MolProbity[54]. All molecular figures were prepared with the program PyMOL[55].

**Quantum chemical calculation**. All residues, retinal, and 19 waters in the proton channel were selected as targets for QM/MM calculations. The initial coordinates of the K state were extrapolated from 7XJC determined in this study, while those of the bR state were from 5ZIL. The protonation states at the crystallization pH (5.6) were estimated with the program PROPKA[56]. Hydrogen atoms were added with the programs PDB2PQR[57] and MAESTRO[58]. The coordinates of the H atoms were optimized under the QM/MM scheme with the program suite GAMESS (US)[59]. The QM region includes Tyr57, Arg82, Asp85, Trp86, Thr89, Trp182, Tyr185, Pro186, Asp212, Lys216, retinal, W401, W402 and W406 (Supplementary Fig. 9a). The QM calculations were performed at the M06-2X/6-31d(d,p) level as recommended for retinal-containing rhodopsins[60]. The MM region was treated with the CHARMM36 force field[61]. At first, the H atoms of water molecules were optimized with the pure MM calculation. Next, only the H atoms of the QM region were optimized by fixing the non-H atoms. Finally, all atoms of the QM region were optimized. The rmsd values between the crystal structure and the optimized structure were 0.13 Å for the K state and 0.07 Å for the ground state, respectively, for non-H atoms of the QM region (Supplementary Fig. 9b, c). As for all non-H atoms of the QM and MM regions, these values were 0.33 Å and 0.27 Å. The NCI plots of the reduced density gradient were calculated from the DFT electron densities using the program NCIPLOT[62]. The potentials for Cε−Nζ and Cδ−Cε bonds were calculated for free $CH_2 = NH-CH_2-CH_2-CH_3$ with Psi4 using HF/6-31 G*[63].

**Statistics and reproducibility**. More than 10 crystals were tested in the diffraction experiment, and data from three crystals (Crystals I−III) were selected for the structure analyses based on resolution. The K and ground state structures from a crystal giving the best resolutions (Crystal I) were described in the main text, while the other two were given in the supplementary materials. The mean values and the

standard deviations for torsion angles of retinal calculated from the three structures for each are shown as hollow circles and error bars (Fig. 2b, Fig. 5b, and Supplementary Fig. 6).

**Reporting summary**. Further information on research design is available in the Nature Portfolio Reporting Summary linked to this article.

## Data availability

The coordinates and structure factors have been deposited in the Protein Data Bank under accession numbers 7XJC (for bR+K), 7XJD (for bR), and 7XJE (for extrapolated K). Source data for figures can be found in Supplementary Data.

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

## Acknowledgements

We thank Messrs. N. Hasegawa and Y. Imai for their help in this work. We also thank the beamline staff at BL41XU of SPring-8 for their help with the data collection experiments (Proposal Nos. 2018B2706, 2019B2718, 2021A2753 to K.T.). This work was supported by the Kyoto University Foundation (to K.T.), the Takeda Science Foundation (to K.T.) and JSPS KAKENHI (No. JP 20H05220 to K.T.). *H. salinarum* was obtained from the RIKEN BRC Cell Bank.

## Author contributions

K.T. supervised the project. S.T. and K.T. designed the experiments. S.T. prepared crystals. S.T., S.N., K.H. and K.T. performed data collections. S.T., S.N., Y.T. and K.T. performed crystallographic analyses. H.-A.D., R.T. and K.T. performed quantum chemical analyses. S.T., S.N., H.-A.D., S.F., K.H. and K.T. discussed the results. S.T. and K.T. wrote the initial draft, and S.N. and H.-A.D. revised the draft. All authors made comments on the draft and consented to submission of the final version.

## Competing interests

The authors declare no competing interests.
