## [Peer Review File · Communications Biology]

Reviewers' comments:

Reviewer #1 (Remarks to the Author):

The work "Detailed analysis of distorted retinal and its interaction with surrounding residues in the K intermediate of bacteriorhodopsin" by Taguchi et al. describes X-ray high-resolution structure of the cryo-trapped K intermediate of bacteriorhodopsin (BR) from *Halobium Salinarum*.

BR is the most studied membrane protein and is the reference in many studies of proton transport mechanisms. Numerous efforts were applied to understand the molecular mechanism of BR function. After the advent of in meso crystallization and the following X-ray high-resolution structures it seemed that the goal will be soon reached. It seemed to be the case. To reach the goal in the case of BR one does not only have to solve the ground but also the intermediate state structures at least to true atomic resolution. In addition, the crystallographic data must be radiation and twinning free, etc. This is the reason why the structures published by different groups were/are controversial and led to diverse proposals of the mechanisms of BR. Only now the scientists started to approach the goal. The K state is not an exception. The published structural data of five different groups are not the same. Some of them differ considerably. The K state is important for the understanding of how the energy of photon is transferred to protein. It is considered that in the K state part of the energy is stored in the disturbed retinal and its nearest environment and then is used by the protein to transport the proton against the membrane electrochemical gradient.

The authors provided one more structure of the K state. They were able to collect crystallographic data to 1.33 Å with moderate occupancy, twinning and quite low radiation damage. I appreciate this very much. The presented data do not coincide with the previous ones, however, in some cases the structure is quite close in some parts to what has been published. Here I have to mention that in general structural rearrangements of the protein in the K state are very small and therefore it is a great challenge to figure out a precise structure of the K state. I find three results of the work of the group of Takeda important. First, the authors provided a high quality crystallographic structure of the K state. Second, they suggested (if I am not mistaken, for the first time) a new explanation of different structures of the K state published by different groups. The authors assumed that different structures correspond to different K sub-states. Third, based on quantum chemical calculations for the K structure, they proposed a relaxation manner to the next L intermediate.

I am of the opinion that publishing this very useful work in *Communications Biology* will be of interest to many scientists. Nevertheless, I would strongly suggest revising the manuscript. My major suggestion is that the differences in the K state structures from different research groups can be explained also (at least in some cases) by high radiation damage, or/and high twinning, or/and insufficient resolution. Also, the authors' explanation of the differences by the different sub-states is interesting but requires a deeper and more concrete analysis.

Minor comments:

Abstract. Line 5. Please, specify a type of X-ray analysis (for instance, instead of "X-ray analysis" write "X-ray crystallographic analysis")

Page 5. Lines 2. Please, comment more why the temperature was lowered to 15K and compare the dose with what was used in all other works on the K state.

Page 5. Lines 2 from the bottom of the page. "...BR and K" should be replaced with "...the BR and K".

Page 6. Line 1. "The root mean square deviation (rmsd) of K in crystal II to K in crystal I was 0.9 Å" – it seems too large. What is the reason?

Page 9. Line 1 of the new paragraph. "...three and two X-ray structures determined with the cryo-trapping or TR-SFX techniques have been reported for the K intermediate." The word "correspondingly" would make the sentence clearer.

Page12. Line 5. "The various crystallographic K structures, which..." More correct: "Some of the various crystallographic K structures, which..."

Page12. Lines 10-11. As I have mentioned "...the change near the SB can be proposed as J → 7XJC (this work) → 1M0K → 6GA6 → 1IXF → 6G7K → 7Z0C → 1QK0 → L." sounds interesting, however, the other reasons for the various structures must at least be mentioned.

Page 13. Line 2 of the second paragraph. "...functional expression" is a misleading part of the sentence.
Page 14. Line 3 of the first paragraph of the "Methods". Please, provide crystallization conditions directly.
Page 15. Paragraph: "Crystallographic analysis" Please, explain how twinning was taken into account.
Fig. 4 "Datils.." is a misprint

Reviewer #2 (Remarks to the Author):

The manuscript by Taguchi et al describes high resolution crystal structures of the light-driven proton pump bacteriorhodopsin (bR), a membrane protein. bR contains a retinal chromophore that absorbs in the blue/green region of the spectrum; the protein is a model system for many other retinal proteins. These are important for either medical reasons, such as visual rhodopsin, or of interest for practical optogenetics applications, such as channel rhodopsin or halorhodopsin. Rhodopsins have rich photocycles, consisting of a number of intermediates with life times ranging from hundreds of femtoseconds to the millisecond range. Upon photon absorption, the retinal chromophore ultimately isomerizes, and the first intermediate formed a few picoseconds after isomerization (in the J intermediate) is the K-intermediate, containing a twisted retinal. This intermediate can be accumulated when illuminating a bR sample kept at cryogenic temperature, providing an experimentally easy access. Accordingly, several structures of the K-intermediate have been determined in the last 20 years. Moreover, with the advent of X-ray free-electron laser, enabling ultrafast time-resolved crystallography at ambient temperature, structures of essentially all reaction intermediates of bR have been determined, including of the K-intermediate.

Taguchi et al describe very high-resolution crystal structures determined by cryocrystallography of bR in the light-adapted dark state and of the K-intermediate. They compare the structures to previously determined ones, describe differences, and try to rationalize these. In principle, the work is very interesting and also important, particularly in view of the recent aforementioned time-resolved studies that used extremely high photoexcitation conditions, compromising the mechanistic insight due to multiphoton excitation.

There are, however, several issues.

1.) Recently a cryogenic study of the K intermediate was published by Borshchevskiy. Despite of the experimental similarities and the very high resolution of both structures of the K-intermediate there are significant differences. It is very surprising that these are not addressed in detail in the manuscript but instead the structure of the L intermediate of Borshchevskiy is discussed. I assume the reason is that the authors have no real idea about the origins of these differences. Among other things they could be due to differences in excitation (no details about the power density is given in either study) or due to local radiation damage (see also Borshchevskiy et al JMB 2011; low dose is necessary but not sufficient for claiming no damage. Lack of damage needs to be shown by spectroscopic analysis of the crystal during data collection). The differences in the two structures and lack of discussion thereof, however, limits the claim of "we report here an accurate X-ray analysis of the K structure". Why is the Taguchi et al structure accurate and the one of Borshchevskiy et al not? Shouldn't this be a major point of the manuscript?

Unfortunately, neither Borshchevskiy et al nor Taguchi et al took optical spectra of the crystals during X-ray data collection. It is thus not clear whether one or both studies suffer from X-ray induced changes affecting the retinal conformation and thus the surrounding residues.

2.) The structures determined by Borshchevskiy and Taguchi et al are unique in their very high resolution and the excellent data quality afforded by conventional synchrotron-based MX. This is in stark contrast to the SFX structures. It would seem that the coordinate error in particular of the Nass Kovacs et al structures is on the order of the described differences (It is not as bad for Nogly et al.) questioning some of the conclusions.

3.) The characteristics of the late and early K intermediates depend on temperature (see for example Dioumaev & Braiman, J. Phys Chem B 1997). It would thus seem that the assignment of the various structures of the K state to K0, KE, KL makes little sense (Page 12). Moreover, it is very likely that a number of these structures are affected by radiation damage, which is detrimental for mechanistic interpretation in case of bR (see e.g. Borshchevskiy et al JMB 2011). It is thus not clear what these structures represent. Analogously, although the SFX structures are free of radiation damage, they most likely affected by multiphoton effects, questioning again what the structures represent.

4.) The authors determine the structure of K intermediate using extrapolated structure factors (presumably this is the structure with code 7XJE). This structure displays much larger differences to the ground state than 7XJC. Why is this? Why is the structure 7XJC used for the comparison with other structures and not 7XJE? This would imply that the authors do not trust the latter? (Is it due to issues with the occupancy of K? One way of determining the value of f (or alpha) is to increase it gradually until features of the ground state start to show up again.

Missing experimental details:

The power density of the sun's radiation on the surface of the earth is approximately 1.4 kW/m². Thus, what is missing for the claim that similar intensity as sunlight was used to generate the K-intermediate (implying biological relevance), is the area of the light spot, and, ideally a number for the power density (mW/mm² or mW/cm²)

Crystals size?

Were the crystals rotated during white and green light illumination? Presumably they were thicker than ~ 6 um, the penetration depth of the exciting 530 nm light
What was the time between light adaption and cryocooling?
It would be an important control to analyze F(red)-F(dark) maps to check whether they are indeed featureless.

Minor issues:

In general, it would help the reader to add the citation to the various PDB codes: 6G7K -> 6G7K(18)

Reference 44: Kabsh -> Kabsch

Page 3: The structural change begins... not true. There are lots of structural changes before the retinal isomerizes.

In one photo reaction cycle -> photon

Page 4: artificial activation -> controlled/ deliberate activation
Most well -> best

Page 5: add "light-adapted" bR crystal was irradiated...

Page 7: changes are limited in -> limited to
bottom W402 increases from -> increases compared to

Page 9: retinal conformation in almost -> in most

Page 14: white halogen lump -> lamp

Reviewer #3 (Remarks to the Author):

The paper investigates the proton pumping mechanism of bacteriorhodopsin through determination of the X-ray structure of the K intermediate, which is the first to occur after isomerization of retinal from all trans to 13 cis. Previous structures of the K intermediate have been reported using both cryotrapping as well as time-resolved serial femtosecond X-ray (TR-SFX) crystallography which differ in the specifics of the retinal conformation and its interactions with the amino acid residues. The current paper is a significant addition to this prior literature and the retinal structure as well as its amino acid interactions are accurately characterized together with quantum chemical calculations.

The major findings are an accurate X-ray analysis of the retinal structure in the K intermediate, where the polyene chain is found to be S-shaped, with characterization of numerous noncovalent interactions within the retinal binding cavity. Work is very carefully performed and thoroughly described. The work is an advance over prior research yet does not significantly alter the current knowledge framework. It represents a useful refinement and comparison to previous synchrotron X-ray cryotrapped structures and TR-SFX structures of the K intermediate. The paper is clear and well-written with excellent figures. It will mainly be of interest to specialists in the field.

Page 5 - The general reader would like to see some proof that indeed it is the K intermediate that is formed under the actual experimental conditions, for example as determined with micro-spectrophotometry. Are the authors able to provide such information?

Page 9-10 - comparison to previous structures using TR-SFX determined under conditions of multiphoton excitation is important and the discussion could be more insightful, e.g. see the paper by D. Miller et al.

Minor comments:

Page 9, line 9 change to "... in most structures..."

Figure 1 - please check gamma and delta symbols

Reviewer #1

Comment 1-1:

My major suggestion is that the differences in the K state structures from different research groups can be explained also (at least in some cases) by high radiation damage, or/and high twinning, or/and insufficient resolution. Also, the authors' explanation of the differences by the different sub-states is interesting but requires a deeper and more concrete analysis.

Our reply:

According to this suggestion, we added a table which shows the differences in experimental conditions, caring of radiation damage, twin ratio, resolution and so on in the Supplementary Table 1 of the revised manuscript. For some structures, explanations for these were added in the text of the revised manuscript (page 10, line 6 from the bottom – page 11, line 7; page 13, lines 9–15).

Comment 1-2:

Abstract. Line 5. Please, specify a type of X-ray analysis (for instance, instead of "X-ray analysis" write "X-ray crystallographic analysis")

Our reply:

We corrected "X-ray analysis" and "X-ray data" to "X-ray crystallographic analysis" and "X-ray diffraction data", respectively.

Comment 1-3:

Page 5. Lines 2. Please, comment more why the temperature was lowered to 15K and compare the dose with what was used in all other works on the K state.

Our reply:

Our previous work had shown that measurements at 15 K resulted in ~2-fold suppression of X-ray damage compared to measurements at 100 K (ref. 25). Consequently, the X-ray diffraction data were measured at 15K in this study. The description was added in the Methods section of the revised manuscript (page 16, lines 16–18). In addition, the dose for this study and other studies were listed in Supplementary Table 1 of the revised manuscript.

Comment 1-4:

Page 5. Lines 2 from the bottom of the page. "...BR and K" should be replaced with "...the BR and K".

Our reply:

The sentence was revised accordingly (page 5 line 4 from the bottom in the revised manuscript).

Comment 1-5:

Page 6. Line 1. “The root mean square deviation (rmsd) of K in crystal II to K in crystal I was 0.9 Å” – it seems too large. What is the reason?

Our reply:

The large rmsd between the K structures of crystal I and II are mainly caused by the difference of shape in the EF loop, and the shrinkage of the c-axis cell constant of crystal II from the others is thought to be the root cause. The description was added in the revised manuscript (page 5, line 3 from the bottom – page 6, line 4). The same can also be said for the bR structure of crystal II.

Comment 1-6:

Page 9. Line 1 of the new paragraph. “...three and two X-ray structures determined with the cryo-trapping or TR-SFX techniques have been reported for the K intermediate.” The word “correspondingly” would make the sentence clearer.

Our reply:

The sentence was modified according to the suggestion (page 9, lines 15–16 in the revised manuscript).

Comment 1-7:

Page 12. Line 5. “The various crystallographic K structures, which...” More correct:” Some of the various crystallographic K structures, which...”

Our reply:

The sentence was corrected according to the comment (page 12, line 2 from the bottom in the revised manuscript).

Comment 1-8:

Page 12. Lines 10-11. As I have mentioned “...the change near the SB can be proposed as J → 7XJC (this work) → 1M0K → 6GA6 → 1IXF → 6G7K → 7Z0C → 1QK0 → L.” sounds interesting, however, the other reasons for the various structures must at least be mentioned.

Our reply:

This comment is related to comment 1-1. We added descriptions as explained in the reply for it.

Comment 1-9:

Page 13. Line 2 of the second paragraph. “...functional expression” is a misleading part of the sentence.

Our reply:

The words “functional expression” were deleted from the sentence in the revised manuscript (page 14, line 13).

Comment 1-10:

Page 14. Line 3 of the first paragraph of the “Methods”. Please, provide crystallization conditions directly.

Our reply:

Crystallization conditions were added in “Methods” of the revised manuscript (page 15, lines 15–20).

Comment 1-11:

Page 15. Paragraph: “Crystallographic analysis” Please, explain how twinning was taken into account.

Our reply:

The description was added in the paragraph of the revised manuscript (page 17, lines 8–3 from the bottom).

Comment 1-12:

Fig. 4 “Datils..” is a misprint

Our reply:

The misprint was corrected in the figure legend of Fig.4 of the revised manuscript.

Reviewer #2

Comment 2-1:

Recently a cryogenic study of the K intermediate was published by Borshchevskiy. Despite of the experimental similarities and the very high resolution of both structures of the K-intermediate there are significant differences. It is very surprising that these are not addressed in detail in the manuscript but instead the structure of the L intermediate of Borshchevskiy is discussed. I assume the reason is that the authors have no real idea about the origins of these differences. Among other things they could be due to differences in excitation (no details about the power density is given in either study) or due to local radiation damage (see also Borshchevskiy et al JMB 2011; low dose is necessary but not sufficient for claiming no damage. Lack of damage needs to be shown by spectroscopic analysis of the crystal during data collection). The differences in the two structures and lack of discussion thereof, however, limits the claim of “we report here an accurate X-ray analysis of the K structure”. Why is the Taguchi et al structure accurate and the one of Borshchevskiy et al not? Shouldn’t this be a major point of the manuscript?

Unfortunately, neither Borshchevskiy et al nor Taguchi et al took optical spectra of the crystals during X-ray data collection. It is thus not clear whether one or both studies suffer from X-ray induced changes affecting the retinal conformation and thus the surrounding residues.

Our reply:

Since large crystals were used for diffraction experiments in the two studies, spectral analysis of the crystals during data collection is virtually impossible. Therefore, we measured the diffraction data of K at an absorption dose of 0.05 MGy, which is lower than the dose limit of 0.15 MGy estimated in our previous paper (ref 25). The fact that X-ray damage is negligible was confirmed indirectly by comparison with data measured in a different order. In response to the comment, the comparison with the K structure of Borshchevskiy et al. has been added to the revised manuscript (page 10, line 4 from the bottom – page 11, line 9). We think that the structural difference is mainly due to the effect of different data collection methods as described in the revised manuscript. However, the resolution of their data is only 1.53 Å, whereas resolutions of the other intermediates reported in the paper are higher (1.0 – 1.22 Å). Furthermore, it should be noted that the crystallographic statistics for the highest resolution shell are insufficient compared to the general case (*e.g.* $I/\sigma(I) = 0.3$, completeness = 39.2%). Therefore, it is possible that the actual accuracy of the analysis may be lower than the apparent resolution. As for the structure of the L intermediate, on the other hand, the resolution is sufficiently high and the crystallographic statistics are sufficient, so we used their L structure for the discussion in our paper. The resolution of our K structure may be comparable to the resolutions of their other intermediates because our criterion is more strict than theirs. Thus, the value of our structures can have a synergistic effect with their L structure. (The K structure is not the main result in their paper.)

Comment 2-2:

The structures determined by Borshchevskiy and Taguchi et al are unique in their very high resolution and the excellent data quality afforded by conventional synchrotron-based MX. This is in stark contrast to the SFX structures. It would seem that the coordinate error in particular of the Nass Kovacs et al structures is on the order of the described differences (It is not as bad for Nogly et al.) questioning some of the conclusions.

Our reply:

The authors' comment was reflected in the revised manuscript (page 13, lines 13 – 15).

Comment 2-3:

The characteristics of the late and early K intermediates depend on temperature (see for example Dioumaev & Braiman, J. Phys Chem B 1997). It would thus seem that the assignment of the various structures of the K state to K0, KE, KL makes little sense (Page 12). Moreover, it is very likely that a number of these structures are affected by radiation damage, which is detrimental for mechanistic interpretation in case of bR (see *e.g.* Borshchevskiy et al JMB 2011). It is thus not clear what these structures represent. Analogously, although the SFX structures are free of radiation damage, they most likely affected by multiphoton effects, questioning again what the structures represent.

Reply:

As commented by the reviewer, each of the K intermediate structures reported so far contains problems. Therefore, careful consideration must be needed when mapping them to their respective sub-states. We added descriptions of the problems associated with each structure in the text, as described in Comments 1-1 and 2-2. Rather than ignoring the problematic structures, we hope to find clues regarding the original structural changes contained therein through structural comparisons. For this reason, we leaved the discussion for the sub-states.

Comment 2-4:

The authors determine the structure of K intermediate using extrapolated structure factors (presumably this is the structure with code 7XJE). This structure displays much larger differences to the ground state than 7XJC. Why is this? Why is the structure 7XJC used for the comparison with other structures and not 7XJE? This would imply that the authors do not trust the latter?

(Is it due to issues with the occupancy of K? One way of determining the value of f (or alpha) is to increase it gradually until features of the ground state start to show up again.

Reply:

Extrapolated structure factors (data for 7XJE in our case) usually contains errors due to the scalar approximation to structure-factor extrapolation (SASFE) as described in ref. 45. So, we used the K structure of 7XJC as the final result in the paper and 7XJE from the extrapolated data were used only as the initial model in the analysis of 7XJC. However, the error is estimated to be only a few percent at most in this case according to a formula derived in ref. 45. Actually, the difference rmsd is 0.33 Å for all protein atoms, which is small enough to conclude that they are nearly identical especially for structures around retinal (Fig. R1a attached in the end of this file).

Comment 2-5:

The power density of the sun's radiation on the surface of the earth is approximately 1.4 kW/m². Thus, what is missing for the claim that similar intensity as sunlight was used to generate the K-intermediate (implying biological relevance), is the area of the light spot, and, ideally a number for the power density (mW/mm² or mW/cm²)

Reply:

The power density of our experiment was added in the revised manuscript (page 15, the first line) and Supplementary Table 2. The value (~0.1 W/cm²) is in the same order of that of sun light (1.4 kW/m² = 0.14 W/cm²).

Comment 2-5:

Crystals size?

Our reply:

The information for the crystal size was added in “Methods” of the revised manuscript (page 16,

line 3) and Supplementary Table 2.

Comment 2-6:

Were the crystals rotated during white and green light illumination? Presumably they were thicker than $\sim 6 \mu\text{m}$, the penetration depth of the exciting 530 nm light

Our reply:

The crystal was rotated during laser irradiation for 5 min (300 sec). This description was added in “Methods” of the revised manuscript (page 16, lines 4 – 5). The optical density at 532 nm of the crystal, which is about $30 \mu\text{m}$ thick, is estimated to be ~ 4 , so that $1/100$ of the light can reach the center of the crystal ($15 \mu\text{m}$ depth). Since P_0 (fraction of non-activated bR) = $\exp(-P \times I)$ is calculated to be ~ 0 in this case (P (photosensitivity of bR) = $10^{-8} \mu\text{m}^2$, I (light intensity) = $(1/100) \times 3 \times 10^9 \times 300 \text{ photons}/\mu\text{m}^2$), this amount of light is sufficient to activate bR molecules at the center of the crystal. The crystal was also irradiated with white light more than 5 min with a white halogen lamp in addition to light from a microscope as additionally described in the revised manuscript (page 15, lines 4-1 from the bottom).

Comment 2-7:

What was the time between light adaption and cryocooling?

Reply:

The crystal was frozen immediately (within ~ 1 min) after light adaptation. The description was added in the revised manuscript (page 15, lines 4-1 from the bottom).

Comment 2-8:

It would be an important control to analyze $F(\text{red})-F(\text{dark})$ maps to check whether they are indeed featureless.

Our reply:

In the measurement of our main data (from crystal I), in order to measure the K intermediate data at as low a dose as possible, we measured the $F(\text{green})$ data as $F(\text{bR}+\text{K})$ at first and then the $F(\text{red})$ as $F(\text{bR})$. For this reason, unfortunately, it is not possible to make the $F(\text{red}) - F(\text{dark})$ map from data of the same crystal. However, in the data measurement from crystal III, $F(\text{dark})$ was measured as $F(\text{bR})$ at first. Since the lattice constants of crystal I and crystal III are almost the same ($R_{\text{iso}} = 9\%$), the structure factors from a single component of the twinned crystals were derived by the method of Pratt *et al.*, (ref. 52) and used to obtain the $F(\text{red}) - F(\text{dark})$ difference Fourier map as the $F_{\text{I}}(\text{bR}) - F_{\text{III}}(\text{bR})$ map (This method is implicated in the SHELX program (ref. 51)). As a result, we could confirm that there is no significant difference electron density in the $F(\text{red}) - F(\text{dark})$ map (Supplementary Fig. 4d–f). On the other hand, the $F_{\text{I}}(\text{bR}+\text{K}) - F_{\text{III}}(\text{bR})$ map generated in the same way clearly shows difference electron density around retinal due to the generation of the K intermediate (Fig. R1b attached in the end of this file), even though the two data are not from the

same crystal. This indicates the validation for the making of the difference Fourier map between different crystals. The featureless result of the $F(\text{red}) - F(\text{dark})$ map is also supported by the superposition of bR of crystal I and bR of crystal III, where no significant differences exist between two structures (Supplementary Fig. 4a–c). The description for the above confirmations were added to the revised manuscript (page 6, lines 5–12 and Supplementary Fig. 4). Although these also indicates the absence of the severe X-ray damage, it was already described for the K structures even in the original manuscript (it is corresponding to page 5 lines 4–3 from the bottom of the revised manuscript).

Comment 2-9:

In general, it would help the reader to add the citation to the various PDB codes: 6G7K -> 6G7K(18)

Our reply:

The citations to the PDB codes were added in the revised manuscript.

Comment 2-10:

Reference 44: Kabsh -> Kabsch

Our reply:

Thank you for pointing out the spelling mistake. The typo was corrected in the revised manuscript.

Comment 2-11:

Page 3: The structural change begins... not true. There are lots of structural changes before the retinal isomerizes.

Our reply:

The sentence was revised to “The light absorption causes the isomerization of a double bond at C13–C14 of retinal from a *trans* to *cis* conformation.” in the revised manuscript (page 3, lines 10–11).

Comment 2-12:

In one photo reaction cycle -> photon

Our reply:

The sentence including the words was revised to “In the reaction cycle of bR, a single proton is transported from the inside to the outside of the plasma membrane.” in the revised manuscript (page 3, lines 14–15).

Comment 2-13:

Page 4: artificial activation -> controlled/ deliberate activation

Our reply:

The word “artificial” was replaced with “deliberate” in the revised manuscript (page 4, line 10).

Comment 2-14:

Most well -> best

Our reply:

The words “most well” were replaced with “best” in the revised manuscript (page 4, line 15).

Comment 2-14:

Page 5: add “light-adapted” bR crystal was irradiated...

Our reply:

The word “light-adapted” was added in the revised manuscript (page 4, line 2 from the bottom).

Comment 2-15:

Page 7: changes are limited in -> limited to

Our reply:

The sentence was corrected in the revised manuscript (page 7, line 15).

Comment 2-16:

bottom W402 increases from -> increases compared to

Our reply:

The word “from” was replaced with “compared to” in the revised manuscript (page 8, line 7).

Comment 2-17:

Page 9: retinal conformation in almost -> in most

Our reply:

The word “almost” was revised to “most” in the revised manuscript (page 9, line 8 from the bottom).

Comment 2-18:

Page 14: white halogen lump -> lamp

Our reply:

The typo was corrected in the revised manuscript (page 15, line 3 from the bottom). Thank you for pointing out the spelling mistakes and grammatical errors.

Reviewer #3 (Remarks to the Author):

Comment 3-1:

Page 5 - The general reader would like to see some proof that indeed it is the K intermediate that is formed under the actual experimental conditions, for example as determined with microspectrophotometry. Are the authors able to provide such information?

Our reply:

Since the thickness of the crystals used in this study was $\sim 30 \mu\text{m}$, the optical density at 532 nm is ~ 4 , and the value at the absorption peak is even larger. This makes spectroscopic measurement very difficult. In addition, microspectrophotometers were not available at the measurement in the beamline BL41XU. In other studies (refs. 15 and 16), different thinner crystals than those used for diffraction data measurements were used for microspectroscopic measurements. However, this does not allow to determine the occupancy of K in the diffraction data. It is also not possible to monitor the progress of X-ray damage during the measurement by the same reason. Therefore, we have followed the established method to produce the K intermediate (ref. 15). The assessment of the X-ray damage was performed by comparison with data measured in a different order, as described in the text (page 5, lines 6–4 from the bottom). We will perform spectroscopic monitoring of crystals in the next study for the controlled generation of the respective sub-states of K.

.

Comment 3-2:

Page 9-10 - comparison to previous structures using TR-SFX determined under conditions of multiphoton excitation is important and the discussion could be more insightful, e.g. see the paper by D. Miller et al.

Our reply:

Inspired by the paper by Miller *et al.*, (ref. 40), the comparison with the TR-SFX structure and the discussion for the relation to the multi-photon process were already described in the last paragraph of “Discussion” section of the original manuscript (corresponding to pages 14–15 of the revised manuscript). The reviewer may be requesting additional discussion, but such a discussion of the specific mechanisms of multiphoton absorption and relaxation effects of tryptophan residues on water molecules is beyond our ability, and we await future developments in the spectroscopic and computational studies. We hope that this study will help in this regard. The citations of the reference and figure (Supplementary Fig. 8d and e) were modified for the reader's understanding in the revised manuscript.

Comment 3-3:

Page 9, line 9 change to "... in most structures..."

Our reply:

The word “almost” was revised to “most” in the revised manuscript (page 9, line 8 from the bottom). (This was also pointed out in comment 2-17).

Comment 3-4:

Figure 1 - please check gamma and delta symbols

Our reply:

The mistake in the label of Fig. 1 was corrected in the revised manuscript (see below). Thank you for your careful check.

The corrected parts (in panel a) are enclosed by red circles.

Figure R1. Figures for the review process. (a) Superimposition of the K structure in 7XJC determined from the bR+K data (magenta) and that in 7XJE determined from the extrapolated data (pink). (b) The $F_I(\text{bR+K}) - F_{III}(\text{bR})$ map. The difference electron densities at the $+3\sigma$ and -3σ levels are shown as green and red meshes, respectively. The ground state structure in crystal III is represented as sticks.

REVIEWERS' COMMENTS:

Reviewer #1 (Remarks to the Author):

My comments were addressed properly

Reviewer #3 (Remarks to the Author):

The authors have adequately addressed all of the referee comments in my view. The proton pumping mechanism of bacteriorhodopsin is investigated by determination of the X-ray structure of the K intermediate, the first to occur after retinal isomerization. Structures of K have been reported using both cryotrapping as well as time-resolved serial femtosecond X-ray (TR-SFX) crystallography. There are differences mentioned in the current revision, but they are not explored with regard to the effects of multi-photon excitations and the difference in the pump laser power densities. All of this is potentially significant.

In the current paper the retinal structure as well as its amino acid interactions are accurately characterized together with quantum chemical calculations and compared to previous synchrotron structures of K. The comparisons of the synchrotron cryo-trapped structures to femtosecond time-resolved structure is deemed to be potentially significant with regard to the effects of the very high-power densities and multiphoton excitations. I would have preferred that the authors had undertaken a more detailed comparison of the cryo-trapped structure with the TR-SFX structures for bacteriorhodopsin. Still, the present paper provides a useful introduction that one can presume will be followed up in more detailed studies. The paper is a useful contribution to the structural literature on bR intermediates and I am happy to support publication in its present form.

List for revisions in the revised manuscript

Comments from Reviewer #2

Comments:

Discussion:

"6G7K and 6Ga6 WERE [not are] determined from TR-SFX data, in which the coordinate error is usually large due to difficulties in measurement and analysis of such data." This statement is not correct. I would either omit this sentence or write "6G7K and 6Ga6 WERE [not are] determined from TR-SFX data using multiphoton excitation. "

Our response:

We deleted this sentence in the revised manuscript according to the comment.

Comment:

Page 16, X-ray diffraction data collection.

crystal size is given in mm^3 , this should be μm^3 The value for the detector distance is missing.

Our response:

We changed mm^3 with μm^3 . The value for the detector distance was added.

Requests in the AIP file

Request:

(Supplementary information)

If you include a title page, please check that the title and author list matches the main manuscript.

Our response:

We added a title page to the supplementary information of the revised version. We checked that the title and author list matches the main manuscript.

Request:

Please provide the values of individual data points for Figures 2b and 5b for the source data file in Excel or text format uploaded as Supplementary Data.

Our response:

The source data for Figure 2b and 5b were provided as Supplementary Data 1 and Supplementary Data 2, respectively, in Excel format.

Request:

(Main text)

Avoid the use of the word “significant” unless referring the results of a statistical test.

Noted ‘significant contribution’ on page 14.

Our response:

The word “significant in the sentence was changed with “great” in the revised manuscript.

Request:

(Display items)

Please show individual data points in Figures 2b and 5b.

Our response:

Figure 2b and 5b were revised according to the request. In addition average values were also shown in the figures. The figure legends for these were revised accordingly.

Request:

Please define error bars in Figure 5b and Supp Fig 6.

In addition, please define the heatmap and units, if applicable, in the panel Figure 6a.

Our response:

We have responded to the requests. The figure legends for these were revised accordingly.

Request:

(Methods)

The Methods should include a separate section titled “Statistics and Reproducibility”

Our response:

The section was added in the revised manuscript.

Request:

Table is present as Table 1. Please check that it fits with the template.

Our response:

We added lines for *B* factors and R.m.s. deviations in Table 1 according to the template.

Request:

(Data Policies)

Please change the ‘Accession codes’ statement to a Data Availability statement.

Our response:

The statement was changed in the revised manuscript.

Other revisions

Revision:

In the course of the revision according to the requests of the AIP file (preparation of the source data for Figure 2b and 5b and adding individual data points in these figures), we noticed that the torsion angles of retinal in the K state in the previous manuscript were not those derived from the final structure deposited to PDB but were from a structure in a mid-way of the analysis. However, the differences between the values and the final values are sufficiently smaller than the errors, and it does not affect the conclusions of the manuscript in any way. So we corrected the values of the torsion angles in the revised manuscript (Page 10, Line 2; Figure 2b; Figure 5b; Supplementary Figure 6; Supplementary Table 4).

The notation “Wat402” was unified with “W402”. (Page 13, Line 5; the legend of Fig. 6).

“K structure” in the legend of Figure 2 was replaced with “K structure (7XJC)” in order to identify the PDB ID like the case of the ground state structure.

The starting point of a dotted line for a hydrogen bond between Asp85 and Thr89 was incorrectly located in Figure 3d of the previous version, and it was corrected.

“Average” in Supplementary Table 4 was replaced with “Mean”.